# Reliability of AlphaFold2 Models in Virtual Drug Screening: A Focus on Selected Class A GPCRs

**DOI:** 10.3390/ijms251810139

**Published:** 2024-09-21

**Authors:** Nada K. Alhumaid, Essam A. Tawfik

**Affiliations:** Advanced Diagnostics and Therapeutics Institute, Health Sector, King Abdulaziz City for Science and Technology (KACST), Riyadh 11442, Saudi Arabia; nkalhumaid@kacst.gov.sa

**Keywords:** AlphaFold2, G protein-coupled receptor, structure validation, molecular docking, virtual screening

## Abstract

Protein three-dimensional (3D) structure prediction is one of the most challenging issues in the field of computational biochemistry, which has overwhelmed scientists for almost half a century. A significant breakthrough in structural biology has been established by developing the artificial intelligence (AI) system AlphaFold2 (AF2). The AF2 system provides a state-of-the-art prediction of protein structures from nearly all known protein sequences with high accuracy. This study examined the reliability of AF2 models compared to the experimental structures in drug discovery, focusing on one of the most common protein drug-targeted classes known as G protein-coupled receptors (GPCRs) class A. A total of 32 representative protein targets were selected, including experimental structures of X-ray crystallographic and Cryo-EM structures and their corresponding AF2 models. The quality of AF2 models was assessed using different structure validation tools, including the pLDDT score, RMSD value, MolProbity score, percentage of Ramachandran favored, QMEAN Z-score, and QMEANDisCo Global. The molecular docking was performed using the Genetic Optimization for Ligand Docking (GOLD) software. The AF2 models’ reliability in virtual drug screening was determined by their ability to predict the ligand binding poses closest to the native binding pose by assessing the Root Mean Square Deviation (RMSD) metric and docking scoring function. The quality of the docking and scoring function was evaluated using the enrichment factor (EF). Furthermore, the capability of using AF2 models in molecular docking to identify hits with key protein–ligand interactions was analyzed. The posing power results showed that the AF2 models successfully predicted ligand binding poses (RMSD < 2 Å). However, they exhibited lower screening power, with average EF values of 2.24, 2.42, and 1.82 for X-ray, Cryo-EM, and AF2 structures, respectively. Moreover, our study revealed that molecular docking using AF2 models can identify competitive inhibitors. In conclusion, this study found that AF2 models provided docking results comparable to experimental structures, particularly for certain GPCR targets, and could potentially significantly impact drug discovery.

## 1. Introduction

Knowledge of the three-dimensional (3D) structure of proteins is crucial for understanding the biological process and addressing different challenges associated with human health through modulating cellular mechanisms and discovering new drugs [1]. Biologists have experimentally determined the structures of more than 200,000 proteins, but this represents a small fraction of the hundreds of millions of known protein sequences [2]. The growing gap between known protein sequences and 3D experimental protein structures has overwhelmed scientists for almost half a century due to the computational complexity of molecular dynamic simulation, the reliance on protein stability, and the challenge of producing sufficiently accurate models of protein physics. The accurate state-of-the-art prediction of protein 3D structures is needed to address this gap and enable large-scale bioinformatics analyses [3].

One of the fundamental grand challenges in biology has reached a new era. Next-generation tools, including AlphaFold2 (AF2) [3], AlphaFill [4], RoseTTAFold [5], and many others, have established a significant breakthrough in structural biology. Establishing these tools has led to the evolution of the landscape of protein structure prediction [6]. Google DeepMind has developed an artificial intelligence (AI) AF2 system that provides a state-of-the-art prediction of protein structures from nearly all known protein sequences with high accuracy [7]. AF2 is thought to have a huge impact on the drug discovery field [1].

From an input of an amino acid sequence of a protein, AF2 used a deep convolutional neural network to provide an output of the 3D structure of the target protein [3,8]. The protein folding that determines the 3D structure of the input sequence is predicted using multiple sequence alignment (MSA), which is constructed from several databases to determine the pattern of correlated mutations. The AF2 AI system is trained on existing data of Protein Data Bank (PDB) structures and capable of predicting the 3D structure of a novel protein using the identified patterns [9,10]. The outstanding performance of AF2 has yielded high-accuracy protein 3D structure predictions that have never been seen in the Critical Assessment of Protein Structure Prediction (CASP) [1,11].

Despite the impressive accuracy of AF2 predictions of protein domain structure, the flexible parts of protein, including the intrinsically disordered protein and loop regions, failed to reach high accuracy and confidence [12,13]. The AI prediction algorithm of AF2 is only able to predict the protein 3D structure without ligands and cofactors. These missing non-protein components are essential for structural integrity and biological function [14]. To overcome this issue in the AF2 system, AlphaFill was created to enrich the AF2 model with ligands and cofactors [4].

A recent paper published in *Nature* announced the release of AlphaFold3 (AF3) in May 2024. Building on the achievements of its previous versions, AlphaFold 3 utilizes improved deep learning methods and expanded datasets to predict protein structures more accurately. Its developments seek to expand the boundaries of structural biology, providing a deeper understanding of protein function and supporting drug discovery and disease research. However, the code for AlphaFold3 will only be available for non-commercial purposes, with restrictions on permitted ligands and covalent modifications [15]. Additionally, using AlphaFold3’s code with any automated system to predict the binding or ligand–protein interaction is not permitted. It is expected that there will be even more remarkable advancements in this technology soon.

The significant breakthrough of AlphaFold has established a new era for G protein-coupled receptor (GPCR) research. GPCRs are an attractive drug target class widely utilized for therapeutic interventions [16,17]. GPCRs are the most prominent membrane protein family containing seven transmembrane α-helical domains connected by alternating intracellular and extracellular loop regions. GPCR functions or signaling impairment has been implicated in various pathophysiological diseases [18]. Based on the amino acid sequences of GPCRs, they are divided into six classes. Still, the human GPCR family has only four main subfamilies (class A, B, C, and F) [19,20]. Among GPCR classes, class A is the largest and the most druggable target class. More than 80% of all GPCRs belong to class A, associated with a wide range of physiological functions [21]. Considering the small fraction of GPCRs that are experimentally solved [22], there is a need to evaluate the AF2 models of GPCRs and investigate their reliability in drug discovery. They have an extensive involvement in versatile biological processes and an implication in various pathophysiological diseases such as central nervous system, cardiovascular, metabolic, and gastrointestinal diseases. Although numerous published studies have compared the performance of AF2 models to experimental structures on a given benchmark that contains some GPCRS, few of them have thoroughly focused on GPCR class A proteins since the introduction of Alphafold2 [22,23,24,25,26,27,28].

Therefore, this research aims to evaluate the reliability of AF2 models in virtual drug screening, explicitly focusing on class A GPCRs. We will compare AF2 models with 32 protein targets determined by X-ray crystallography and Cryo-electron microscopy (cryo-EM). The quality of the AF2 models will be assessed using various structure validation tools, including the pLDDT score, RMSD value, MolProbity score, percentage of Ramachandran favored, QMEAN Z-score, and QMEANDisCo Global. The reliability of AF2 models in virtual drug screening will be determined based on their ability to predict ligand binding poses closest to the crystal binding pose for selected GPCR class A drug targets. Additionally, we will assess the screening effectiveness of molecular docking and the potential of using AF2 models in molecular docking to identify hits with key protein–ligand interactions against the chosen protein targets. The outcomes of this comprehensive assessment study will offer insights into the reliability of AF2 models, specifically for the virtual screening of small-molecule inhibitors.

## 2. Results

To investigate the reliability of AF2 models compared to experimental structures in drug discovery, a benchmark set of 32 representative protein targets from the class A GPCR family belonging to 20 subfamilies were selected, including Serotonin, Angiotensin, Bombesin, Cannabinoid, Cholecystokinin, Leukotriene, Chemokine, Dopamine, Endothelin, Free fatty acid, Ghrelin, Lysophospholipid (LPA and S1P), Acetylcholine (muscarinic), Tachykinin, Orexin, P2Y, Neuropeptide Y, Adrenoceptors, Opioid, and Orphan (Figure 1). Seventy-six ligands were selected to be docked into all the PDB structures.

### 2.1. Quality of Class A GPCR AlphaFold2 Models

Table 1 shows the model’s quality assessment of each of the selected class A GPCRs. The pLDDT score, RMSD value, MolProbity score, percentage of Ramachandran favored, QMEAN Z-score, and QMEANDisCo Global are displayed. Most AF2 models were computed with confidence (pLDDT values ≥ 70). Only the pLDDT metric of the Acetylcholine M3 receptor indicates that it was computationally predicted with low confidence. All RMSD values of AF2 backbone atoms are highly relative to the corresponding crystal structure, particularly the binding site backbone’s RMSD values. However, the visual inspection of AF2 models shows that some protein structures adopt different disordered conformations compared to their corresponding X-ray structure. As shown in Figure 2, the extracellular loops of A Orphans (GPR52), Somatostatin (SST2), Chemokine (CCR2), and Opioid (δ) AF2 models were pulled toward the inside of the protein’s binding site.

The MolProbity score is one package of a structure stereochemistry analysis that combines the clash score, percentage of bad side chain rotamers, and percentage of Ramachandran not favored. The MolProbity scores of AF2 models are generally low, which reflects good quality at both the global and local levels. However, the MolProbity scores of Serotonins (5-HT2B and 5-HT5A), Dopamine (D2), and Acetylcholine (M3) are relatively higher than 2, which might indicate inferior quality. The Ramachandran plot demonstrated more than 80% of backbone dihedral angles (ϕ and ψ) of residues in AF2 structures in their energetically favored regions. The MolProbity score and percentage of Ramachandran favored of the selected class A GPCRs analyze the overall AF2 model geometry, indicating that AF2 models were computed with reasonable structural stereochemical quality.

The QMEAN Z-scoring function was utilized to investigate whether the AF2 model is of comparable quality to high-resolution X-ray crystallographic structures by estimating the degree of nativeness of the structural features of AF2 models. QMEAN Z-scores around zero indicate a high degree of nativeness compared to high-resolution X-ray structures of similar size; however, most AF2 models of the selected proteins exhibit Z-scores < −4.0, indicating low-quality models. The QMEANDisCo score estimates structure model quality by combining the single-model composite score (QMEAN) with a consensus-based distance constraint (DisCo) to derive global (entire protein structure) and local per-residue absolute quality. The QMEANDisCo Global score is the mean per-residue quality score, and the provided error estimate is calculated according to QMEANDisCo Global scores of a large set of models of similar size to the input. Most AF2 models of the selected proteins have QMEANDisCo Global scores higher than 0.60. Overall, AF2 models of the selected class A GPCRs exhibit good-quality models.

### 2.2. Molecular Docking Scores of the Selected Class A GPCRs

The results of molecular docking for the crystal ligand of each receptor are depicted in Figure 3. The crystal ligand of each receptor was docked into the experimental structures (X-ray and Cryo-EM) as well as the corresponding AF2 models to evaluate the accuracy of the docking methodology in predicting the crystal ligand pose using AF2 models. Generally, the docking results of the crystal ligands were better with the X-ray structures.

However, AF2 results were comparable to Cryo-EM results for 5-HT2B, 5-HT5A, BB2, CB2, BLT1, CCR5, D2, Ghrelin, P2Y1, and κ. AF2 models exhibited improved docking results for the crystal ligands compared to Cryo-EM in 5-HT2C, CCK1, FFA1, M4, and Y2. For β1, the AF2 model yielded higher docking results than the X-ray structure but lower than the Cryo-EM structure. Similar docking results for AF2 models with all experimental structures were noted in NK1 and OX2, and comparable docking results for the AF2 models with X-ray structures in M4. Inferior docking results of AF2 models compared to experimental structures were observed in D3, ETB, GPR52, LPA1, M3, P2Y12, S1P1, SST2, α2A, and μ. Among the 32 GPCR AF2 models selected, CCR2 and δ exhibited the worst docking results compared to all experimental structures.

The results of the molecular docking for the constructed ligand library (consisting of 76 active ligands) are displayed in Figure 4. The box and whisker plot illustrates the distribution of docking results for experimental structures (X-ray and Cryo-EM) and their corresponding AF2 models across quartiles. In most cases, AF2 models showed comparable PLP-fitness scores to the experimental structures. These include 5-HT2B, 5-HT2C, 5-HT5A, AT1, BB2, CB2, CCK1, BLT1, CCR5, D2, D3, ETB, FFA1, Ghrelin, LPA1, M4, NK1, OX2, P2Y1, P2Y12, S1P1, SST2, Y2, β1, κ, and μ. The most notable differences were observed for receptors CB1, CCR2, M3, α2A, and δ, where the docking results of AF2 models were significantly lower than the corresponding experimental structures. However, Cryo-EM also demonstrated inferior results for M3. Additionally, there were frequent extreme outliers in AF2 models, particularly for CB1, FFA1, and M4.

### 2.3. Posing Accuracy of Docked Ligands

The accuracy of ligand positioning was assessed to evaluate the ability of the docking algorithm to predict the ligand binding poses closest to the crystal binding pose. The proportion of conformations with RMSD values less than 2.0 Å between the docking pose and the experimental conformation of the crystal ligand was plotted and shown in Figure 5. The X-ray structure demonstrated superior posing accuracy compared to Cryo-EM structures and AF2 models in predicting the poses of docked crystal ligands (Figure 5A). X-ray structures showed a ~20% increase in posing accuracy compared to Cryo-EM and AF2, resulting in 80% posing accuracy. However, when the constructed ligand library (76 active ligands) was used as input, the performance of AF2 models was comparable to or slightly better than the X-ray structure in predicting a successful pose but slightly lower than Cryo-EM structures. The posing accuracy for the entire constructed library of active ligands (Figure 5B) was approximately 40% for the experimental structures (X-ray and Cryo-EM) and their corresponding AF2 models.

Furthermore, an evaluation of the top-scored poses revealed that AF2 models outperformed both X-ray and Cryo-EM structures in terms of posing accuracy for the docked active ligands. Figure 6A–C illustrate the success rates (RMSD between the top-scored pose and the crystal pose lower than predefined thresholds) for the top-scored poses of the constructed ligand library docked into X-ray structures (Figure 6A), Cryo-EM structures (Figure 6B), and AF2 models (Figure 6C). The AF2 models achieved a success rate of 47%, outperforming both X-ray and Cryo-EM structures, which had success rates of 39% and 42%, respectively. In addition to the higher success rate, the failure rate for AF2 models was 34%, with an additional 19% of the poses classified as marginal failures (RMSD between 2.0 Å and 3.0 Å). These results suggest that AF2 models demonstrate superior posing power compared to the experimental structures.

Since the size and flexibility of ligands might affect the resulting posing accuracies of ligands docked into experimental structures and AF2 models, the success rates based on the number of rotatable bonds of ligands were assessed. Figure 6D represents the magnitude of success rates between the number of rotatable bonds of ligands and the protein structure utilized for molecular docking. Generally, ligands with fewer rotatable bonds had higher success rates, but there was no strong correlation observed. AF2 models showed comparable success rates compared to X-ray and Cryo-EM structures in successfully predicting the poses of ligands regardless of the number of rotatable bonds. Ligands with eight rotatable bonds had the lowest success rates, around 4–6%, for both experimental structures and AF2 models. Collectively, the heatmap in Figure 6D shows that AF2 models are comparable to experimental structures at handling flexible ligands.

### 2.4. Evaluation of Screening Power

The virtual screening of actual positive ligands and decoys was performed using three structural methodologies: X-ray crystallography, Cryo-EM, and AF2 predicted models. Table 2 shows that the performance of each method was evaluated using three metrics: the enrichment factor (EF), hit rate (HR%), and percentage of correctly classified ligands in the top 5% of compounds across the selected class A GPCR receptors. The mean EF values for X-ray structures, Cryo-EM structures, and AF2 models were 2.24, 2.42, and 1.82, respectively (Table 2). Cryo-EM structures demonstrated the highest overall enrichment across receptors, outperforming both X-ray structures and AF2 models. X-ray structures performed moderately well, with a slightly lower EF compared to Cryo-EM. In contrast, AF2 models showed the lowest EF, indicating that their performance in docking-based enrichment was inferior to the experimental structures.

The mean hit rates followed a similar trend. Cryo-EM structures achieved the highest mean HR% of 31.98%, while X-ray structures displayed a mean HR% of 29.58%. AF2 models showed a lower mean HR% of 23.96%, reflecting their reduced ability to identify correct hits compared to both experimental methods. The HR% values for AF2 models displayed greater variability across different receptors, as indicated by the higher standard deviation (12.46) compared to Cryo-EM (10.30) and X-ray structures (11.22).

In terms of correctly classified ligands, Cryo-EM structures again demonstrated superior performance, with a mean of 12.62%, compared to 11.68% for X-ray structures and 9.46% for AF2 models. The lower classification accuracy of AF2 models is consistent with their reduced performance in EF and HR%. Notably, the variability in the percentage of correctly classified ligands was also higher for AF2 models, as reflected in their standard deviation of 4.92, compared to X-ray structures (4.43) and Cryo-EM structures (4.06).

Across individual receptors, Cryo-EM structures consistently performed well, showing particularly strong results for receptors such as BB2 (EF = 4.04, HR% = 53.33%) and CB1 (EF = 3.54, HR% = 46.67%). Despite the worsened average EF value for AF2 models, AF2 models outperformed Cryo-EM structures in several receptors including CCK1, BLT1, CCR5, GPR52, M4, Y2, and α2A. Furthermore, the number of AF2 models showed higher EF values than the X-ray structures including BB2, CCK1, D3, Ghrelin, GPR52, P2Y1, P2Y12, β1, and μ. Equal EF values between X-ray structures and AF2 models were observed in CCR5, LPA1, and M4. Consequently, these AF2 models showed higher percentages of HR and correctly classified ligands. AF2 models performed best for the Y2 receptor (EF = 4.29, HR% = 56.67%, correctly classified ligands = 22.37%) and the Ghrelin receptor (EF = 3.54, HR% = 46.67%, correctly classified ligands = 18.42%). For some receptors, including CB1, CCR2, and δ, AF2 models displayed particularly poor results, with much lower EF and HR% values compared to the other methods.

### 2.5. Analysis of Ligand Competitive Inhibition

The potential of using AF2 models in molecular docking to identify competitive inhibitors was thoroughly analyzed. The protein–ligand interactions for the top-scored ligands docked against AF2 models yielded promising results. Figure 7 presents the binding sites of top-scored ligands (magenta sticks) for selected targets in AF2 models, along with their 2D protein–ligand interaction diagrams. For comparison, the AF2 models are shown in cyan, while the corresponding X-ray structures are depicted in green, illustrating a generally good superposition of the AF2 models within the binding site relative to the experimental X-ray structures.

The docking pose of the top-scored ligand for the Cannabinoid receptor (CB2) is shown in Figure 7A. The protein–ligand interactions (Figure 7B) reveal that the docked ligand forms four π–π stacking interactions with Phe281, His95, and Phe91, along with three π–cation interactions with Phe183 and Trp194. Additionally, a hydrogen bond was observed between the ligand and Ser90, and a salt bridge interaction formed between the ligand’s ammonium group and the carboxylate side chain of Asp101.

For the Dopamine receptor (D3), the docking pose of the top-scored ligand is illustrated in Figure 7C. The corresponding interaction diagram (Figure 7D) reveals two salt bridges between the ligand’s ammonium groups and the carboxylate side chains of Asp110 and Glu90. Additional interactions include π–cation bonding with His349 and π–π stacking with the phenyl group of Tyr365.

In Tachykinin (NK1), the docking pose of the top-scored ligand is shown in Figure 7E. As shown in Figure 7F, the docked ligand is present with π–cation interaction with the phenyl group of Tyr287 and π–π stacking with the phenyl group of His197 residue. Halogen bonding was also present between the docked ligand and the Asn89 residue.

In Adrenoceptors (β1), the docking pose of the top-scored ligand is shown in Figure 7G. The protein–ligand interaction of the docked ligand revealed a salt bridge between the ammonium group and the side chain carboxylate group of the Asp121 residue, as shown in Figure 7H. Further interactions involved π–π stacking with the phenyl groups of Phe307 and Phe201 residues.

A complete list of the top-scored ligands for each protein target, identified using AF2 models, is provided in Appendix A. These ligands demonstrated superior protein–ligand interactions compared to the corresponding crystal ligands. These results suggest that molecular docking with AF2 models is not only feasible but also effective in identifying competitive inhibitors, highlighting its potential for drug discovery applications.

## 3. Discussion

Closing the gap between known protein sequences and the 3D experimental protein structures is only possible by predicting the structures of millions of proteins [6]. The development of the AF2 AI system eases this challenge by making state-of-the-art predictions of the 3D structures of proteins from their amino acid sequences [29]. Despite the breakthrough in structural biology established by AF2 development, some obstacles are associated with the AF2 predictions. The AF2 AI system is currently trained on exiting the X-ray crystallographic structures available in the PDB [7]. AF2 has difficulty in predicting the intrinsically disordered protein and loop regions. Moreover, the AF2 system cannot predict the protein folding defects due to point mutation, or identify the apo and holo forms of the protein [30].

Assessments of the confidence measures and the quality of AF2 protein structures are critical before docking. In this current study, the selected class A GPCR AF2 models were built with high confidence, and the quality of structures was good in terms of structure stereochemistry analyses (Table 1). It should be noted that the AF2 AI algorithm relies on multiple sequence alignments and cannot predict the novel protein structures [30]. All the selected class A GPCRs have multiple experimental structures, including high-resolution X-ray crystallographic structures. Consequently, the AF2 AI system can compute these models with high confidence. It has been particularly indicated that AF2 modeled GPCRs more accurately than template-based modeling [31].

A closer look into each AF2 model’s quality assessment revealed that the seven transmembrane α-helixes of GPCRs were built with high confidence. In contrast, the alternating intracellular and extracellular loop regions were built with lower model confidence—the results of pLDDT metrics aligned with QMEANDisCo metrics (see Appendix A). Mainly, the alignment of results was returned by pLDDT and QMEANDisCo metrics for each of the selected class A GPCRs (Table 1). Both of these measures assess the local accuracy of the predicted protein regions rather than the entire protein structure to drive global accuracy.

To assess the reliability of the AF2 models in virtual drug screening, ligand-bound X-ray crystallographic structures and the corresponding Cryo-EM structures, resolved at a high resolution of ≤3.0 Å, were selected for a comparative analysis of molecular docking performance. Since AF2 proteins were modeled without ligands, co-factors, metals, ions, and water [7], the experimental structures were stripped from non-protein components except the crystal-bound ligands in the active site. This approach ensured that the docking comparison between AF2 models and experimental structures (X-ray and Cryo-EM) was performed on an equal basis.

To standardize the molecular docking methodology, we used the crystal-bound ligand of each protein to define the binding site for both the AF2 models and the experimental structures. This allowed us to evaluate the docking performance of the AF2 models fairly and compare it directly to that of the experimentally determined structures. Our methodology enabled a more accurate assessment of the AF2 model’s ability to identify ligand binding interactions under conditions comparable to those of X-ray and Cryo-EM structures. This provides a robust benchmark for virtual drug screening applications.

The molecular docking and virtual drug screening were performed using GOLD software, which has benefited from continuous enhancement and benchmarking for posing and sampling powers [32,33,34,35]. The ChemPLP scoring function was chosen since recent validation tests indicated that PLP fitness is more effective for pose prediction and virtual drug screening than the other scoring functions [36]. Initially, the bound ligand of each selected X-ray protein was extracted and used as a reference for the crystal binding pose. Thereby, the reliability of AF2 models in virtual drug screening was assessed by the ability to predict the ligand binding poses closest to the crystal binding pose using AF2 models to benchmark the performance of virtual drug screening toward the selected GPCR class A drug targets. The molecular docking was made by using the crystal ligand of each protein and the constructed compound library of 76 active ligands with high inhibitory activities (Ki) against class A GPCRs. Each AF2 model underwent evaluation using molecular docking scoring with the PLP-fitness function, and the posing accuracy of each resulting docking pose was assessed using the RMSD metric. Docking results of AF2 models were further investigated to identify ligands with competitive inhibition against the selected targets.

The results of the molecular docking analysis for class A GPCRs provide a comprehensive comparison between docking scores obtained from experimental structures (X-ray and Cryo-EM) and AF2 models. Overall, X-ray structures yielded the best docking results for crystal ligands (Figure 3). However, AF2 models demonstrated notable promise, often performing comparably to experimental structures and, in some cases, surpassing them. The docking results revealed that AF2 models exhibited docking scores comparable to Cryo-EM structures for receptors such as 5-HT2B, 5-HT5A, BB2, CB2, BLT1, CCR5, D2, Ghrelin, P2Y1, and κ. In certain cases, AF2 models even outperformed Cryo-EM, as observed for receptors like 5-HT2C, CCK1, FFA1, M4, and Y2, highlighting their potential in virtual screening applications. These findings suggest that AF2 models can serve as reliable alternatives to experimental structures for specific GPCR targets, especially when high-resolution experimental data are unavailable.

When assessing the performance of AF2 models with the constructed ligand library of 76 active ligands (Figure 4), AF2 models produced results comparable to both X-ray and Cryo-EM structures for most receptors. These include 5-HT2B, 5-HT2C, 5-HT5A, AT1, BB2, CB2, CCK1, BLT1, CCR5, D2, D3, ETB, FFA1, Ghrelin, LPA1, M4, NK1, OX2, P2Y1, P2Y12, S1P1, SST2, Y2, β1, κ, and μ. This reinforces the potential of AF2 models for high-throughput virtual screening, as they can produce docking scores that align closely with experimental structures in most cases.

To ensure a reliable and unbiased assessment of pose prediction accuracy, the RMSDs between the docked and crystal ligand crystalized pose were calculated. Overall, if the crystal ligand of each receptor were used as an input for molecular docking, the posing accuracy using the X-ray structure was higher than in AF2 models (Figure 5A). If the constructed ligand library was used as input for molecular docking, AF2 models showed comparable results in posing accuracy to the experimental structures (Figure 5B). This suggests that with a larger dataset, the AF2 models have the potential to outperform the experimental structure, looking into how successful the docking methodology is in correctly predicting the ligand binding poses closest to the crystal ligand pose using AF2 models, particularly for the top-scored active ligands (Figure 6A,C).

The number of rotatable bonds of a ligand has a critical influence on ligand flexibility, which makes the ligand undergo significant conformational changes upon binding that are not captured adequately in the docking algorithms. Inadequate conformational sampling performance during docking simulation will eventually lead to higher RMSD values, particularly when the number of rotatable bonds of ligands is higher than 20 [34,37,38,39,40]. Consequently, the magnitude of success rates between the number of rotatable bonds of ligands and the protein structure utilized for molecular docking was assessed. No strong correlation was observed between the number of rotatable bonds and the success rates of pose predictions (Figure 6D). This is probably because only small molecules with few rotatable bonds in the constructed ligand library were included in this study. Nevertheless, AF2 models performed similarly to X-ray and Cryo-EM structures in accurately predicting the ligand poses regardless of the ligand’s number of rotatable bonds.

In this study, we further evaluated the performance of molecular docking by analyzing its screening power across 32 GPCR class A drug targets (Table 2). The screening power was assessed using the enrichment factor, a crucial measure for determining how effective the docking algorithm is in identifying actual positive ligands from a pool of potential candidates. The enrichment factor is calculated by comparing the proportion of actual positives among the top-ranked ligands to the proportion of actual positives expected by random selection. A higher enrichment factor signifies that the scoring function prioritizes correct positive ligands and distinguishes them from decoys.

Our study observed a variability in AF2 enrichment performance across different receptors. While AF2 performed exceptionally well for certain receptors, such as Ghrelin and CCR5, which nearly matched the results of the experimental structures, it performed poorly for others, such as CB1, CCR2, and δ. This inconsistency emphasizes that AF2 models cannot compete with the experimental structures in virtual drug screening but can provide valuable performance. The inferiority of AF2 performance is due to their structural model limitations as they lack non-protein components critical for the integrity of the structure and biological function. Additionally, they adopt different disordered conformations compared to their corresponding X-ray structures (Figure 2). Furthermore, there are slight variations in AF2 models compared to the desired biological conformations, especially in terms of backbone RMSD, particularly at the backbone level of ligand binding residues (Table 1). Therefore, model modifications increase the chances of docking success. Nevertheless, a generative AI model aims to accurately predict a protein’s 3D structure to accelerate research in nearly all fields of biology and ease the discovery of novel drugs where experimental protein structures are unavailable rather than replacing them.

The investigation of top-scored ligands identified through molecular docking using AF2 models demonstrated that AF2 is capable of identifying ligands with competitive inhibition potential. The majority of top-scored ligands for each protein target exhibited stronger protein–ligand interactions compared to their corresponding crystal ligands, as shown in Appendix A. These ligands not only achieved higher docking scores but also formed a broader array of interactions with the target proteins, including hydrogen bonding, π–π stacking, π–cation interactions, and salt bridges, surpassing the interactions observed with crystal ligands. While some AF2 models showed slightly inferior docking performance compared to X-ray structures, as noted in Figure 3 and Figure 4, they were still effective in identifying hits with competitive inhibition. This suggests that even when AF2 models do not fully capture the protein binding site as accurately as experimental structures, they can still generate viable predictions that can guide early-stage drug discovery. These results emphasize the potential of AF2 models as valuable tools for identifying promising ligands, especially when experimental structures are unavailable, while also highlighting the need for continued refinement to enhance their docking accuracy. The applicability of the AF2-derived models in identifying novel inhibitors for drug discovery was examined previously, and researchers successfully identified a new inhibitor against a novel target called CDK20, with a Kd value of 9.2 ± 0.5 μM, without an experimental structure and within one month [41]. Another study used the AF2 model and successfully identified a hit against the HDAC11 protein with an IC_50_ value of 3.5 µM [42].

The reliability of crude AF2 models for virtual drug screening is questionable without post-modeling refinement [43]. Slight variations in the AF2 computed side chains would highly impact molecular docking and virtual drug screening [23]. AF2 still needs further improvement to reach docking performance accuracy comparable to the X-ray crystallographic structure [44]. Previous studies have demonstrated that proper refining of AF2 models hugely impacts molecular docking [23]. Refinement of AF2-derived models using the induced-fit docking method (IFD-MD) would significantly enhance the enrichment performance [27]. Another study has presented that refining the multistate AF2-derived models using physics-based tools could improve the performance of the kinetic and thermodynamic data and would help to achieve a level of accuracy comparable to the experimental structures [45]. Also, it has been suggested that stripping the regions with low confidence from predicted AF2 models and enhancing the side chain flexibility will improve the docking outcomes [26].

## 4. Materials and Methods 

### 4.1. Construction of Ligand Library

A total of 76 ligands were downloaded from the Zinc20 Database (https://zinc.docking.org/; accessed in 21 February 2024) in ready-to-dock, three-dimensional formats [46]. The ligands were selected based on their known inhibitory activities (Ki) against class A GPCRs reported in the ChEMBL33 Database (https://www.ebi.ac.uk/chembl/; accessed in 23 February 2024) [47,48]. Each ligand shows high affinity to one or more G protein-coupled receptors with a pKi value greater than 6 (see Appendix A). Appendix A shows the physiochemical properties of the 76 active ligands included in the docking set. All selected ligands were combined in one mol2 file format to ease the molecular docking. GPCR decoys were downloaded from the DUD-E database (https://dude.docking.org/; accessed in 25 February 2024) [49].

### 4.2. Preparation of Receptors

The 32 class A GPCRs used in this study (see Appendix A) were retrieved from the PDB (https://www.rcsb.org/; accessed in 15 September 2024) [50], and the corresponding AF2 models for each receptor were retrieved from the AlphaFold Protein Structure Database (https://www.alphafold.ebi.ac.uk/; accessed in 17 September 2024) [7] using the UniProt accession number [51]. The selected crystallographic and Cryo-EM structures were solved at a resolution ≤ 3.0 Å. Initially, the protein structures were prepared using PyMOL (The PyMOL Molecular Graphics System, Version 2.5.4 Schrödinger LLC, New York, NY, USA) to match their corresponding AF2 models by removing the crystal water molecules, metals, and cofactors despite the awareness of their importance in ligand binding [52,53,54]. Missing amino acids were added to the selected experimental structures.

### 4.3. Assessment of AlphaFold2 Structures

The predicted Local Distance Difference Test (pLDDT) is reported in the AlphaFold Protein Structure Database (https://www.alphafold.ebi.ac.uk/; accessed in 10 March 2024). The model confidence of the computed structure provides a per-residue confidence score (pLDDT) ranging between 0 and 100. Values of ≥70 indicate regions that are computationally predicted with high confidence, while values of <70 generally indicate regions that are predicted with low confidence [7]. The degree of structural overlap between the X-ray structures and their corresponding AF models using backbone atoms was measured by RMSD (i.e., Root Mean Square Deviation) using PyMOL. Alignments with RMSD equal to ≤2 Å are reasonable alignments. The quality of AF models was assessed using the SWISS-MODEL server (https://swissmodel.expasy.org/; accessed in 20 March 2024) to analyze stereochemistry by MolProbity [55] and a Ramachandran plot [56] and estimate the model quality by QMEANDisCo [57]. MolProbity determines the quality of the model at the global and local levels; MolProbity scores should be as low as possible. The Ramachandran plot was used to predict the percentage of backbone dihedral angles (ϕ and ψ) of residues in protein structure in their energetically favored regions. The QMEANDisCo Global scores above 0.60 indicate good-quality models. The Qualitative Model Energy Analysis (QMEAN) Z-scores around zero indicate native-like structures, and Z-scores < −4.0 indicate low-quality models [58].

### 4.4. Overview of Molecular Docking

The present study investigated molecular docking with Genetic Optimization for Ligand Docking (GOLD) software (Version 2023.1.0) from the Cambridge Crystallographic Data Centre (CCDC, Cambridge, UK). Each protein experimental structure and its corresponding AF models were prepared using the GOLD set-up wizard with the default setting. Since the AlphaFold2 models were built without bound ligands, all AF2 models were aligned to their corresponding protein crystal structure to define the binding site of AF2 models and facilitate the docking calculations. Also, the Cryo-EM structures’ binding sites were defined based on their corresponding protein crystal structure to ensure a fair comparison. Before docking, all hydrogen atoms were calculated and added, and the ligand was extracted for all PDB structures. Protonation and tautomeric states of histidine residues were assigned. The docking site for each PDB structure was defined using the default GOLD cavity detection algorithm and the protein region within 6.0 angstrom (Å) of the extracted ligand. Each ligand is flexibly docked to its associated protein structure binding site [32,59]. GOLD allows some flexibility in the protein side chains in hydrogen bonding while the rest of the protein structure remains fixed. GOLD uses a genetic algorithm (GA) to modify and optimize the dihedrals of protein OH groups and NH_3_^+^ groups, the dihedrals of ligand rotatable bonds, the geometries of ligand rings by flipping ring corners, and the mappings of the fitting points, which refer to the position of the ligand in the binding site [33]. By default, ten GA runs were performed for each PDB structure to optimize the fitness scores for each flexibly docked ligand. When determining the number of genetic operations, GOLD considers both ligand flexibility and the volume of the binding site, balancing efficiency with processing time [36,60]. By default, GOLD terminates the number of GA early as soon as the top three scoring solutions are within 1.5 Å RMSD of each other [36].

### 4.5. Assessment Methods

#### 4.5.1. GOLD’s ChemPLP Scoring Function

The resulting docked poses were ranked by the default GOLD scoring function ChemPLP, which exceeded other GOLD scoring functions in benchmarking studies in terms of speed and accuracy. The scoring function of GOLD’s ChemPLP evaluates how efficiently a ligand binds to protein targets by considering chemical properties (hydrogen bonding, electrostatic interactions, and hydrophobic effects) and the spatial arrangement of the ligand within the binding site. It scores ligand poses based on energy terms, including van der Waals forces and hydrogen bonding, with lower scores indicating better binding predictions. This function optimizes ligand conformations and selects the most promising candidates for further experimental validation. ChemPLP overall fitness scores are positive numbers, with higher scores indicating better performance and greater stability of the receptor–ligand complex [32,59].

#### 4.5.2. Root Mean Square Deviation (RMSD)

This study assessed the accuracy of docking poses by calculating the Root Mean Square Deviations (RMSDs) of the docked ligands, in addition to evaluating the docking scoring function. RMSD measures the average distance between the atoms of the docked ligand and the corresponding atoms in the crystal structure of the ligand. A docking pose is considered successful if the RMSD is within 2.0 Å of the crystal ligand’s experimental conformation; otherwise, it is deemed a failure. This threshold is widely used in docking studies to ensure that the predicted poses closely match the experimentally determined ligand conformations, reflecting both the precision of the docking process and the reliability of the predicted binding mode [32,59,61,62,63]. While RMSD provides a quantitative measure of structural similarity, its interpretation can be influenced by size-dependent factors. For instance, an RMSD < 2 Å indicates better performance for large molecules than for small molecules (five or fewer rotatable bonds). Therefore, RMSDs between 2 and 3 Å have been annotated as “Fail-Marginal” [64]. All RMSD values were calculated using the PyMol script.

#### 4.5.3. Enrichment Factor (EF)

The docking and scoring function’s quality was assessed using the enrichment factor. This critical parameter indicates the likelihood of identifying an actual positive ligand within the top 5% of compounds based on the docking results compared to random selection. The *EF* is defined as
EFx=HitsxNxHitstotalNtotal
where *Hits_x_* represents the number of actual positive ligands present in a subset *x* (top 5%) of the docked library, *N_x_* is the number of compounds in subset *x* (top 5%), *Hits_total_* is the total number of actual positive ligands within the entire chemical library, and *N_total_* is the total number of compounds including actual positive ligands and decoys. High *EF* values indicate better docking and scoring function quality. Apart from the EF calculation, two additional parameters were calculated to evaluate the docking and scoring function, including the hit rate (HR), which is defined as the percentage of actual positive ligands present in the top 5% of compounds $\left(\frac{{Hits}_{x}}{N_{x}}\times 100\right)$, and the rate of correctly classified ligands in the total number of actual positive ligands $\left(\frac{{Hits}_{x}}{{Hits}_{total}}\times 100\right)$ [65].

### 4.6. Ligand Competitive Inhibition Analysis

The docking results for 76 active ligands against AF2 models will be further analyzed to identify potential hits with competitive inhibition against the selected GPCR proteins. The top-scored ligand, which has a higher PLP-fitness score than the crystal ligand, will be chosen. Then, the 2D interactions of the ligand will be illustrated using Maestro (Version 2024-3, Schrödinger, New York, NY, USA).

## 5. Conclusions

In virtual drug screening, an ideal 3D protein structure should predict the binding pose of ligands and identify key protein–ligand interactions to aid in the discovery of new inhibitors for a given protein target. The AF2 revolution has brought a new era to the drug discovery field. Given the notable roles of the GPCR protein family in various pathophysiological functions and the need to predict the 3D structures of known GPCR sequences that have not been experimentally resolved yet, the reliability of AF2 models in virtual drug screening was examined. This study reports the comprehensive assessment of AF2 for molecular docking scoring, posing accuracy, and screening power using the GOLD software, which has benefited from continuous enhancement and benchmarking for compound sampling. Overall, AF2 models showed competitive performance in molecular docking, particularly for certain GPCR targets, often achieving docking scores similar to experimental structures and, in some cases, outperforming them. However, limitations were evident in specific receptors where AF2 models struggled to reproduce the consistently high screening power seen in experimental structures. However, AF2 succeeded in predicting ligand binding poses closest to the crystal binding pose. Importantly, AF2 succeeded in identifying key protein–ligand interactions necessary for competitive inhibition, demonstrating its potential as a tool for virtual drug screening. While AF2 models generally showed slightly inferior screening power compared to experimental structures, their ability to accurately predict binding poses makes them a reasonable option for virtual drug screening when experimental structures are unavailable. AlphaFold3 with its further enhancements is aimed for to improve the reliability of its models for virtual drug screening.

## Figures and Tables

**Figure 1 ijms-25-10139-f001:**
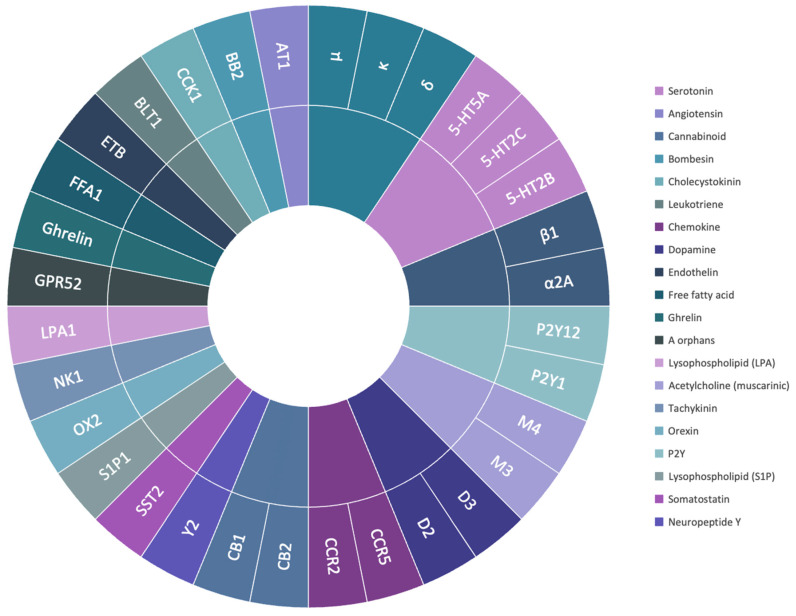
The biological hierarchy of class A GPCRs selected for this study. The hierarchy diagram includes the 20 protein subfamilies and the protein’s IUPHAR codes.

**Figure 2 ijms-25-10139-f002:**
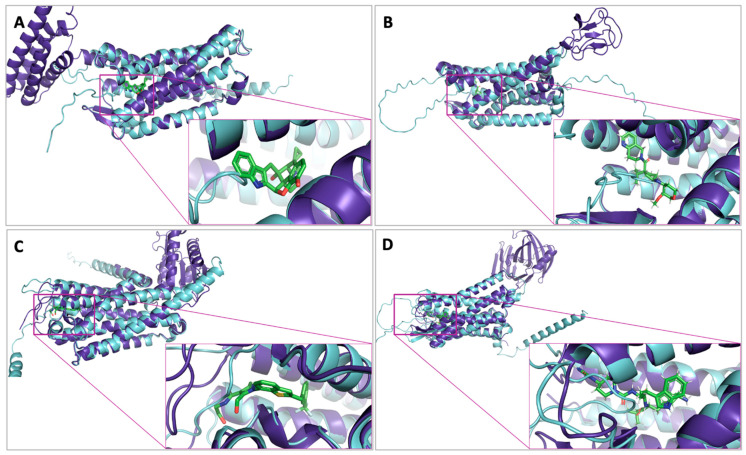
The superposition of X-ray structure (purple-blue color) and the bound ligand (green color) with their corresponding AF2 model (cyan color). (**A**) The binding site of Opioid (δ); (**B**) the binding site of Chemokine (CCR2); (**C**) the binding site of A Orphans (GPR52); and (**D**) the binding site of Somatostatin (SST2).

**Figure 3 ijms-25-10139-f003:**
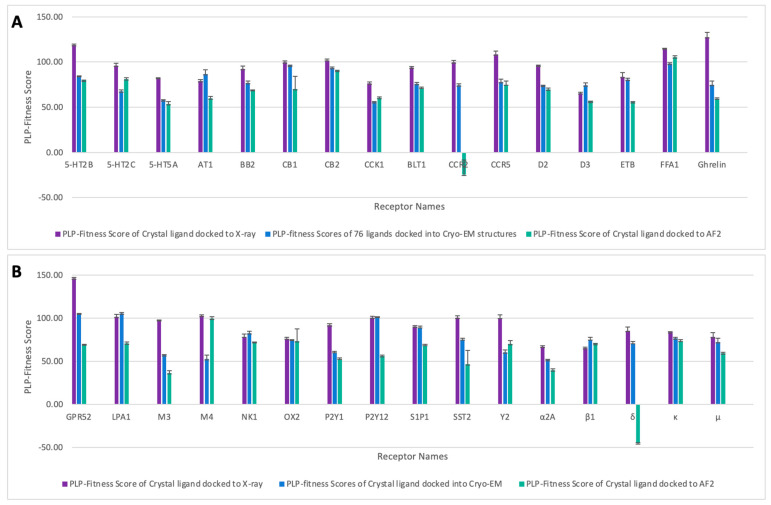
The docking results of the crystal ligand of each receptor docked into the experimental structures (X-ray and Cryo-EM), and the corresponding AF2 models. The results of the selected 32 receptors were divided into two figures, (**A**,**B**), for clarity. The error bars represent the standard deviations.

**Figure 4 ijms-25-10139-f004:**
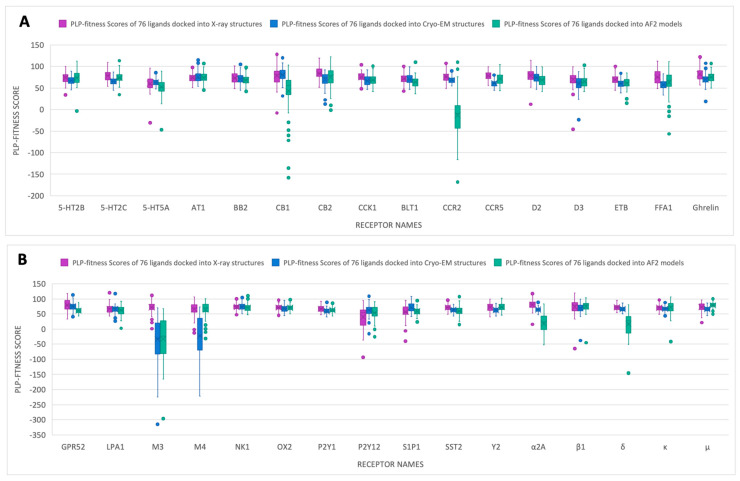
The docking results of 76 ligands docked into the experimental structures (X-ray and Cryo-EM), and the corresponding AF2 models. The results of the selected 32 receptors were divided into two figures, (**A**,**B**), for clarity. All color dots are considered outliers.

**Figure 5 ijms-25-10139-f005:**
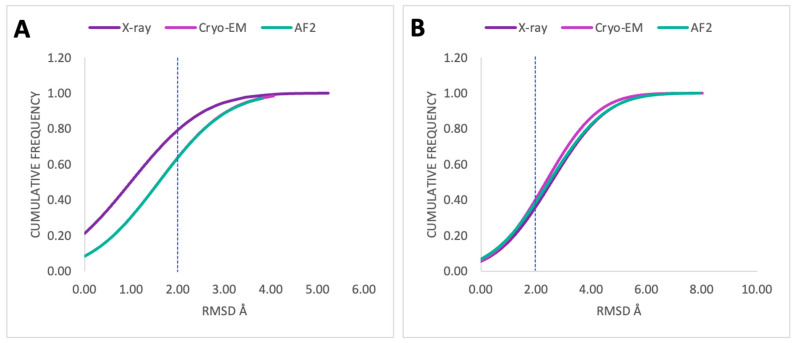
Cumulative distribution plots for docking of protein–ligand complexes. (**A**) When crystal ligands of each receptor were used as input; and (**B**) when the 76 active ligands were used as input. The dotted lines indicate a 2.0 Å RMSD cutoff.

**Figure 6 ijms-25-10139-f006:**
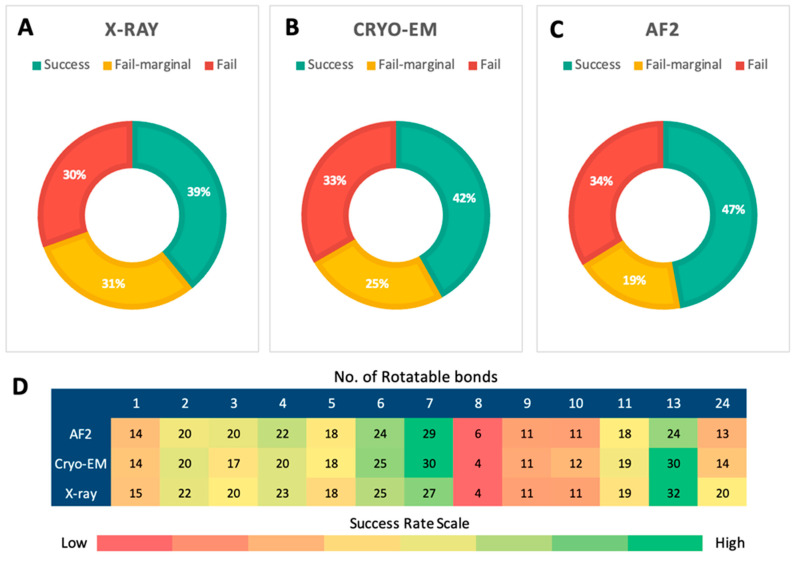
Posing accuracy of the 76 docked active ligands. (**A**) The success rate (%) for the top-scored active ligands docked into X-ray structures; (**B**) docked into Cryo-EM structures; (**C**) docked into the corresponding AF2 models. (**D**) The heatmap of docking success rates for the 76 active ligands with different numbers of rotatable bonds. A docking pose is considered successful if RMSD between the docking pose and the experimental conformation of the crystal ligand is <2.0 Å. Otherwise, a docking pose is considered to have failed. However, an RMSD of 2–3 Å is annotated as ‘Fail-marginal’.

**Figure 7 ijms-25-10139-f007:**
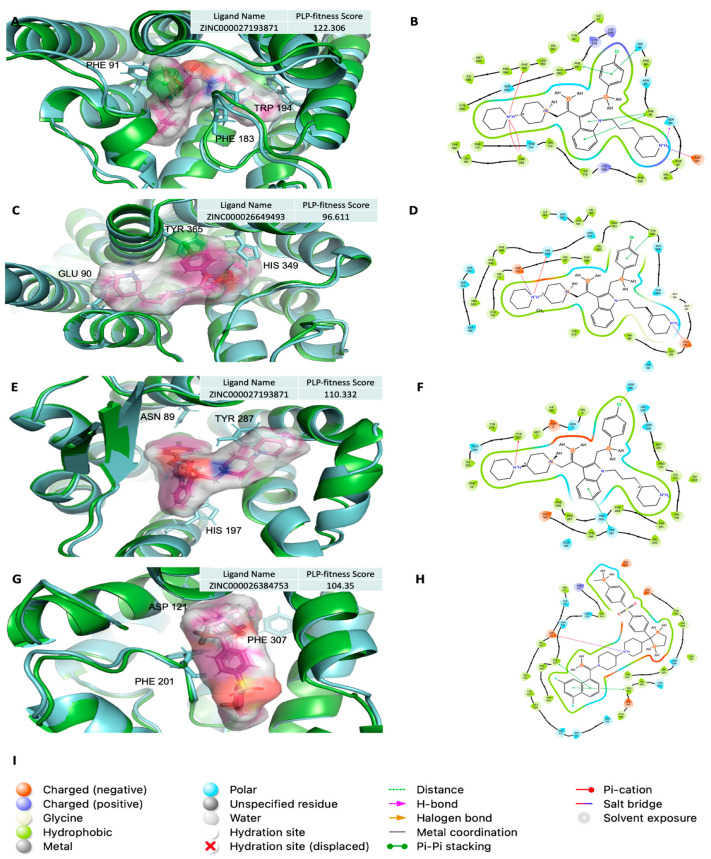
The AF2 binding site of top-scored ligands for selected targets, along with their 2D protein–ligand interaction. AF2 models are displayed in **cyan**, and the corresponding X-ray structure is displayed in **green** for comparison. (**A**,**B**) The binding of the top-scored ligand against Cannabinoid (CB2); (**C**,**D**) the binding of the top-scored ligand against Dopamine (D3); (**E**,**F**) the binding of the top-scored ligand against Tachykinin (NK1); (**G**,**H**) the binding of the top-scored ligand against Adrenoceptors (β1); (**I**) represent the 2D protein-ligand interactions that explain (**B**,**D**,**F**,**H**).

**Table 1 ijms-25-10139-t001:** Assessment of class A GPCR AF2 structure quality.

Receptor	pLDDT (Global) Score	Backbone RMSD (Å)	Binding Site RMSD (Å)	MolProbity Score	Ramachandran Favored (%)	QMEAN Z-Score	QMEANDisCo Global
**5-HT2B**	71.74	0.93	0.45	2.09	88.10	−6.49	0.59 ± 0.05
**5-HT2C**	73.53	0.52	0.44	1.78	87.28	−6.15	0.58 ± 0.05
**5-HT5A**	78.89	1.11	0.66	2.09	87.61	−4.39	0.64 ± 0.05
**AT1**	82.04	1.07	0.48	1.55	91.04	−3.33	0.67 ± 0.05
**BB2**	78.92	1.15	0.90	1.97	90.05	−5.67	0.61 ± 0.05
**CB1**	71.66	1.97	1.17	1.67	90.21	−5.08	0.59 ± 0.05
**CB2**	82.48	0.87	0.38	1.69	93.30	−4.08	0.70 ± 0.05
**CCK1**	78.54	0.81	0.75	1.89	89.20	−4.96	0.61 ± 0.05
**BLT1**	82.53	0.88	0.47	1.77	91.14	−4.91	0.68 ± 0.05
**CCR2**	78.28	0.63	0.50	1.66	86.83	−5.72	0.64 ± 0.05
**CCR5**	85.29	0.77	0.69	1.42	92.57	−4.46	0.70 ± 0.05
**D2**	72.41	0.81	0.52	2.18	87.76	−5.07	0.54 ± 0.05
**D3**	75.64	0.50	0.30	1.80	91.96	−4.52	0.61 ± 0.05
**ETB**	75.52	1.56	0.54	1.90	90.45	−5.27	0.66 ± 0.05
**FFA1**	89.45	0.55	0.37	1.44	94.97	−3.29	0.77 ± 0.05
**Ghrelin**	81.36	1.54	0.59	1.41	95.05	−2.12	0.69 ± 0.05
**GPR52**	81.82	0.65	0.54	1.38	93.59	−3.74	0.65 ± 0.05
**LPA1**	84.50	0.37	0.27	1.69	90.33	−4.97	0.70 ± 0.05
**M3**	67.64	1.12	0.95	2.29	82.11	−7.07	0.51 ± 0.06
**M4**	76.23	0.42	0.24	1.77	88.68	−6.16	0.54 ± 0.05
**NK1**	78.25	0.46	0.31	1.74	90.62	−4.65	0.66 ± 0.05
**OX2**	78.27	0.53	0.27	1.72	92.76	−3.57	0.65 ± 0.06
**P2Y1**	86.18	0.64	0.79	1.34	91.37	−4.30	0.72 ± 0.05
**P2Y12**	85.04	0.79	0.45	1.84	91.47	−4.13	0.70 ± 0.05
**S1P1**	81.67	0.48	0.31	1.69	91.05	−4.90	0.64 ± 0.05
**SST2**	81.56	0.91	0.83	1.98	91.55	−3.98	0.67 ± 0.05
**Y2**	82.42	0.78	0.77	1.66	91.29	−4.17	0.66 ± 0.06
**α2A**	72.11	0.85	0.49	1.61	84.45	−6.13	0.53 ± 0.06
**β1**	74.01	0.59	0.28	1.56	89.40	−5.96	0.57 ± 0.05
**δ**	80.40	0.45	0.24	1.30	94.05	−2.83	0.69 ± 0.05
**κ**	80.07	0.66	0.43	1.51	92.33	−3.88	0.64 ± 0.05
**μ**	77.67	0.96	0.58	1.73	90.40	−3.99	0.64 ± 0.06

**Table 2 ijms-25-10139-t002:** Assessment of the screening power. CI, confidence interval; EF, enrichment factor; HR, hit rate; SD, standard deviation.

Receptor	X-ray Structures			Cryo-EM Structures			AlphaFold2 Models		
	EF	HR%	Correctly Classified Ligands%	EF	HR%	Correctly Classified Ligands%	EF	HR%	Correctly Classified Ligands%
**5-HT2B**	2.53	33.33	13.16	3.28	43.33	17.11	2.02	26.67	10.53
**5-HT2C**	2.27	30.00	11.84	2.53	33.33	13.16	2.02	26.67	10.53
**5-HT5A**	1.52	20.00	7.89	1.26	16.67	6.58	1.26	16.67	6.58
**AT1**	2.78	36.67	14.47	2.02	26.67	10.53	1.77	23.33	9.21
**BB2**	1.77	23.33	9.21	4.04	53.33	21.05	2.78	36.67	14.47
**CB1**	2.78	36.67	14.47	3.54	46.67	18.42	0.76	10.00	3.95
**CB2**	2.27	30.00	11.84	2.53	33.33	13.16	1.01	13.33	5.26
**CCK1**	2.27	30.00	11.84	2.53	33.33	13.16	2.78	36.67	14.47
**BLT1**	3.03	40.00	15.79	1.77	23.33	9.21	2.27	30.00	11.84
**CCR2**	2.53	33.33	13.16	2.53	33.33	13.16	0.00	0.00	0.00
**CCR5**	2.53	33.33	13.16	2.02	26.67	10.53	2.53	33.33	13.16
**D2**	2.27	30.00	11.84	3.79	50.00	19.74	2.02	26.67	10.53
**D3**	1.01	13.33	5.26	1.26	16.67	6.58	1.26	16.67	6.58
**ETB**	1.52	20.00	7.89	3.54	46.67	18.42	1.26	16.67	6.58
**FFA1**	2.78	36.67	14.47	2.78	36.67	14.47	1.26	16.67	6.58
**Ghrelin**	2.02	26.67	10.53	2.27	30.00	11.84	3.54	46.67	18.42
**GPR52**	1.77	23.33	9.21	1.77	23.33	9.21	2.02	26.67	10.53
**LPA1**	1.77	23.33	9.21	2.53	33.33	13.16	1.77	23.33	9.21
**M3**	2.27	30.00	11.84	1.52	20.00	7.89	1.01	13.33	5.26
**M4**	3.28	43.33	17.11	1.01	13.33	5.26	3.28	43.33	17.11
**NK1**	3.28	43.33	17.11	3.28	43.33	17.11	1.26	16.67	6.58
**OX2**	3.03	40.00	15.79	2.53	33.33	13.16	2.53	33.33	13.16
**P2Y1**	1.01	13.33	5.26	2.78	36.67	14.47	2.27	30.00	11.84
**P2Y12**	0.25	3.33	1.32	2.02	26.67	10.53	1.77	23.33	9.21
**S1P1**	1.52	20.00	7.89	1.77	23.33	9.21	1.01	13.33	5.26
**SST2**	3.54	46.67	18.42	2.78	36.67	14.47	1.01	13.33	5.26
**Y2**	4.55	60.00	23.68	2.78	36.67	14.47	4.29	56.67	22.37
**α2A**	2.02	26.67	10.53	1.01	13.33	5.26	1.26	16.67	6.58
**β1**	1.52	20.00	7.89	3.03	40.00	15.79	2.78	36.67	14.47
**δ**	2.53	33.33	13.16	2.02	26.67	10.53	0.25	3.33	1.32
**κ**	2.27	30.00	11.84	2.53	33.33	13.16	1.01	13.33	5.26
**μ**	1.26	16.67	6.58	2.53	33.33	13.16	2.02	26.67	10.53
**Mean**	2.24	29.58	11.68	2.42	31.98	12.62	1.82	23.96	9.46
**SD**	0.85	11.22	4.43	0.78	10.30	4.06	0.94	12.46	4.92
**Lower 95% CI**	1.95	25.69	10.14	2.15	28.41	11.22	1.49	19.64	7.75
**Upper 95% CI**	2.54	33.47	13.21	2.69	35.55	14.03	2.14	28.27	11.16

## Data Availability

The authors confirm that the data supporting the findings of this study are available within the article.

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
