# Peer review of "Reliability of AlphaFold2 Models in Virtual Drug Screening: A Focus on Selected Class A GPCRs"

_ijms, 2024, doi:10.3390/ijms251810139_

Round 1

Reviewer 1 Report

Comments and Suggestions for Authors

The submitted manuscript presents the results of theoretical analysis aimed to evaluate the reliability of AlphaFold2 models in virtual drug screening by focusing on GPCRs class A. In lines 83-86  the Authors perfectly justifies my recommendation to reject this work. “Although numerous published studies have compared the performance of AF2 models to experimental structures on a given benchmark that contains some GPCRS, few of them has thoroughly focused on GPCR Class A proteins since the introduction of Alphafold2”. In other words, the aim of this work has been already achieved at least a few times, therefore, after critical analysis of works [22-28] I truly believe that the level of novelty presented in the submitted manuscript is simply very low. I mean, the study design and presentation style are OK, but this is simply not enough, as the presented conclusions simply confirm what already has been confirmed quite a few times. As alternative, I suggest other MDPI journals such as Applied Biosciences , BioMedInformatics, Future Pharmacology .

Other major issues:

The submitted manuscript is a little out of date, as the authors have used the AlphaFold2 while the improved version AlphaFold3 is available and now no one uses AlphaFold2 anymore. Therefore, the results are of little importance.

Line 14 “and determined by X-ray crystallography” suggests that the Authors have performed the experimental studies on 58 proteins, which is not true, since the work is purely theoretical

Line 15, “Using different structure validation tools” such as? Please list them here

Line 38, here, it should be explained why it is so relatively eyas to get the primary structure while very hard to get the tertiary one

Table 1, this is also a serious drawback. The Authors focus on the protein as a whole, measuring the RMSD of the whole protein. This would be important if the Authors have performed i.e. MD simulations. Docking is a very LOCAL process, therefore, the Authors should focus on the analysis of the binding sites in order to explain the differences in the obtain results upon choosing the structure.

Also, I don’t know why the Authors have used those particular structures of the studied proteins, from the RCSB database. I mean, the experimental X-ray structures of most of the studied proteins were deposited in the database more than once. Instead of focusing on the number of receptors, the Authors should limit the number of receptors and check how the choice of the (experimental) structure affects the results, comparing to the structure from AlphaFold. This may drastically change the discussion and conclusions.

Figure 3, how the error bars were obtained?

Minor comments:

Line 8, I’d rather say „computational biolochemistry” as protein structure prediction requires knowledge from chemistry, biology, physics and informatics

Line 99, it should be “Zinc20 Database”

Table 2 should be edited as right now it is poorly formatted. I mean, it is impossible to read the data such as mean, SD, etc.

Author Response

We attached the response to reviewer 1. Thank you

Reviewer 2 Report

Comments and Suggestions for Authors

The authors used AI to analyze protein structure,  particularly GPCRs Class A. The rationale is adequate; however, it can be improved by analyzing ligand competitive inhibition in some cases. The methodology is generally adequate, although the interpretation of the crystal structure of different ligands was compared partially; the authors did only a partial analysis with similar compounds in which the kinetics of binding and enthalpy should have been used more efficiently. The discussion on the model refers mainly to some of the events analyzed, but not all the possible interactions. The authors should analyze the limitations of the models in Figures 3 and 4, as some points of the crystal structure differ from those of the AI analysis. It is inferred then that the model may require modifications.  

Comments on the Quality of English Language

Minor grammatical mistakes were encountered.

Author Response

We thank the reviewer for their thoughtful and helpful comments. We respond to each point in the revised manuscript. Their suggestions have undoubtedly improved the paper, for which we are very grateful. The new sections in the revised manuscript are highlighted in blue font.

Also, the revised manuscript was proofread.

Round 2

Reviewer 1 Report

Comments and Suggestions for Authors

While I appreciate the answers of the Authors, there is still one major issue that really bothers me. I mean, I can turn a blind eye to the fact that similar works have been reported multiple times before, but please treat the comment below seriously.

My comment: The experimental X-ray structures of most of the studied proteins were deposited in the database more than once. Instead of focusing on the number of receptors, the Authors should limit the number of receptors and check how the choice of the (experimental) structure affects the results, comparing to the structure from AlphaFold. This may drastically change the discussion and conclusions.

Authors Response: The criteria for selection were based on the availability of crystallographic ligand-bound structures, which were solved at a resolution ≤ 3.0 Å. If multiple structures were found, the structure with the best resolution was selected.

My comment in the second review round: It must be checked and discussed how the choice of the structure affects the results. After all, it may turn out that the results exp. Vs AlphaFold are less distinct than exp.”structureA” vs exp”structureB”.

Author Response

Reviewer 1: It must be checked and discussed how the choice of the structure affects the results. After all, it may turn out that the results exp. Vs AlphaFold are less distinct than exp.”structureA” vs exp”structureB”.

Response: Thank you for your comment. Based on your suggestion, we included an additional experimental structure in our study, modified the manuscript, and analyzed all the data accordingly. Please note that due to limitations in structure availability and our selection criteria, the number of receptors was restricted to 32.

Briefly, we analyzed 32 Cryo-EM structures by conducting molecular docking for each receptor's crystal ligand, 76 active ligands, and a combination of active ligands and decoys. We then assessed docking scores, posing accuracy, and screening power (EF) after integrating Cryo-EM structures. This approach helped us evaluate the reliability of AF2 models compared to two experimental structures (X-ray and Cryo-EM), thereby improving the overall validity of our findings.

Reviewer 2 Report

Comments and Suggestions for Authors

The authors made some changes in the manuscript, but the changes were only in parts of the text. The ligand docking interaction was added, but it is still insufficient. I asked for more input on the equation and the analysis, which is still preliminary. 

Author Response

Reviewer 2: The authors made some changes in the manuscript, but the changes were only in parts of the text. The ligand docking interaction was added, but it is still insufficient. I asked for more input on the equation and the analysis, which is still preliminary. 

Response: We appreciate your comment and apologize for the confusion. Based on your suggestions, we have made several changes to the manuscript and more inputs were added, which we hope meet the expectations.

Please note that we have included an additional experimental structure in our study, but due to our selection criteria and limited availability of structures, we could only include 32 receptors. We analyzed 32 Cryo-EM structures by conducting molecular docking for each receptor's crystal ligand, 76 active ligands, and a combination of active ligands and decoys. We then assessed docking scores, posing accuracy, and screening power (EF) after integrating Cryo-EM structures. This approach helped us evaluate the reliability of AF2 models compared to two experimental structures (X-ray and Cryo-EM), thereby improving the overall validity of our findings. Per your recommendations, we have revised the paper and conducted a thorough analysis of all the events.

Round 3

Reviewer 1 Report

Comments and Suggestions for Authors

The Authors have revised and improved their work. This version can be accepted for publication.

Reviewer 2 Report

Comments and Suggestions for Authors

The manuscript has been improved. I am glad that the authors made the requested changes. Now it is suitable for publication